# Encrusted Uretero-Pyelitis Caused by *Corynebacterium urealyticum*: Case Report and Literature Review

**DOI:** 10.3390/diagnostics12092239

**Published:** 2022-09-16

**Authors:** Andrei Valentin Rusmir, Ionut Andrei Paunescu, Sandra Martis, Silviu Latcu, Dorin Novacescu, Claudia Ramona Bardan, Flaviu Bob, Monica Licker, Mircea Botoca, Alin Cumpanas, Razvan Bardan

**Affiliations:** 1Department of Urology, “Pius Brinzeu” Clinical Emergency County Hospital, 300736 Timisoara, Romania; 2Department of Urology, “Victor Babes” University of Medicine and Pharmacy, 300041 Timisoara, Romania; 3Department of Hematology, Clinical Emergency City Hospital, 300723 Timisoara, Romania; 4Department of Nephrology, “Pius Brinzeu” Clinical Emergency County Hospital, 300736 Timisoara, Romania; 5Department of Nephrology, “Victor Babes” University of Medicine and Pharmacy, 300041 Timisoara, Romania; 6Clinical Laboratory, “Pius Brinzeu” Clinical Emergency County Hospital, 300736 Timisoara, Romania; 7Department of Microbiology, Multidisciplinary Research Center on Antimicrobial Resistance, “Victor Babes” University of Medicine and Pharmacy, 300041 Timisoara, Romania

**Keywords:** encrusted uretero-pyelitis, *Corynebacterium urealyticum*

## Abstract

We report the case of a 70-year-old female patient with solitary functioning left kidney and encrusted uretero-pyelitis caused by *Corynebacterium urealyticum*, which was treated by antibiotic therapy and oral acidification with L-methionine. We review the literature for similarly reported cases.

## 1. Background

Encrusted uretero-pyelitis (EUP) is characterized by the presence of calcifications on the wall of the pelvicalyceal system and ureter, usually associated with urinary tract infection caused by *Corynebacterium urealyticum*. It is a very rare and serious disease, first reported by Morales et al. in 1992 [1]. EUP may be associated with encrusted cystitis, which has an identical pathophysiology. Predisposing factors are the presence in the urinary tract of urea splitting bacteria responsible for alkaline urine, a history of urological procedures or pre-existing mucosal lesions, and immunosuppression, which are found in most patients. In many cases, EUP is related to a hospital-acquired infection. The specific aspect of the calcifications on CT scan can help to diagnose the disease [1,2]. The majority of *Corynebacterium urealyticum* strains are multi-drug resistant, but with universal sensitivity to vancomycin and teicoplanin [3]. The treatment of EUP consists of antibiotic therapy with glycopeptides and urine acidification [1,2].

We report the case of a female patient with encrusted uretero-pyelitis caused by *Corynebacterium urealyticum*, with an unfavorable outcome.

## 2. Case Report

### 2.1. Presentation

A 70-year-old female patient, with right atrophic kidney and chronic kidney disease (CKD) stage 3b diagnosed two years before, was admitted to the emergency room with hypogastric pain, urinary frequency, dysuria, persistent macroscopic hematuria, and a mildly altered general condition two years after diagnosis. She had a history of recurrent urinary tract infections (>3 episodes/year, mostly with *E. Coli*) and had undergone an endoscopic bladder biopsy for a suspicion of calcified bladder tumor two years before, for which the pathology revealed changes suggestive of chronic inflammation. Her medical history included Parkinson’s disease (under therapy with rasagiline), stage II essential hypertension (treated with candesartan), bilateral operated pertrochanteric fracture, and severe obesity (BMI = 40.4 kg/m^2^).

### 2.2. Diagnosis

At hospital admission, the clinical examination revealed skin pallor and pain at the percussion of the left lumbar region. Blood tests revealed: CKD stage G5 KDIGO (serum creatinine = 5.7 mg/dL, eGFR = 8 mL/min/1.73 m^2^), mild hyperkalemia (5.9 mmol/L), hypoalbuminemia (2.4 g/dL), and severe hypochromic, microcytic anemia (hemoglobin = 5.5 g/dL, mean corpuscular volume = 74.8 Fl, mean corpuscular hemoglobin concentration = 29.6 g/dL). Urinalysis revealed an alkaline urine (pH = 8.6), low density (1009), proteinuria, and gross hematuria. Standard urine culture was negative.

The peripheral smear showed anisocytosis with microcytes, anisochromia with marked hypochromia, anulocytes, and rare ovalocytes. Based on these results, the hematological evaluation established the diagnosis of severe, multifactorial, hypochromic microcytic anemia. The most probable cause of the anemia was the chronic iron deficiency due to infection (the serum ferritin level was 5 µg/L), which was associated with the renal impairment and subsequent erythropoietin deficiency.

Emergency abdominal and pelvic non-contrast CT scan showed multiple thin left ureteral, pyelic, and calyceal calcifications on the surface of the urothelium, which are specific to encrusted uretero-pyelitis, with a maximum density of 510 Hounsfield units and with a mild dilation of the upper left urinary tract, right kidney hypoplasia, and enlarged retroperitoneal lymph nodes (Figure 1 and Figure 2).

Based on the imaging and laboratory findings, the diagnosis established in the Department of Urology (together with the nephrologist) was of encrusted uretero-pyelitis with obstructive nephropathy and acute kidney injury AKIN 3/CKD G5, CKD-MBD (mineral bone disease) (iPTH = 205.6 pg/mL, serum phosphate = 5.5 mg/dL).

The next step of the investigation included a urethro-cystoscopy, which revealed cloudy urine and a diffuse erythematous swelling of the bladder mucosa, characteristic of chronic cystitis, with no apparent bladder calcifications. Thereafter, during the same operating theatre session under general anesthesia, we inserted a catheter into the left ureter orifice and performed contrast retrograde ureteropyelography, which showed non-homogenous opacification of the collecting system due to the presence of multiple parietal calcifications at the level of the left pelvicalyceal system and ureter. Unfortunately, during this intervention, we did not collect a selective urine culture from the left collecting system, which could have been useful for the diagnosis.

Considering the initial negative urine culture, the high values of the urinary pH, and the presence of the uretero-pelvic calcifications, we thought about a possible infection with *Corynebacterium urealyticum*. Three days after hospital admission, a new urine sample was collected. The cytobacteriological examination of the urine (CBEU) revealed >10^5^ CFU/mL *Corynebacterium urealyticum*. Urine culture was performed on Columbia blood agar and chromogenic media, with a 48 h incubation period at 35–37 °C; identification was performed using matrix-assisted laser desorption/ionization—time-of-flight mass spectrometry (MALDI-TOF, Brucker Daltonics, Billerica, MA, USA). Disk diffusion antimicrobial susceptibility testing using the breakpoints of the European Committee on Antimicrobial Susceptibility Testing (EUCAST) showed susceptibility to fluoroquinolones and glycopeptides.

### 2.3. Therapy

Following the retrograde ureteropyelography, we inserted a left double-J ureteric stent, with the intention of facilitating the urinary drainage from the left kidney, considering that the opposite right kidney was hypoplastic (Figure 3).

Initial antibiotic therapy was performed with intravenous ciprofloxacin 100 mg bid for five days; when the infection with *Corynebacterium urealyticum* was confirmed, considering the lack of clinical improvement from ciprofloxacin, we switched to antibiotic therapy with intravenous vancomycin (15 mg/kg/day). Moreover, urine acidification therapy was started with oral L-methionine at 2 g/day.

Four units of packed red blood cells were given to correct the severe anemia, while the hyperkalemia was corrected by administering calcium gluconate and furosemide, following the recommendations of the nephrologist.

### 2.4. Hospital Discharge

The patient was discharged after fourteen days, with the following lab results: serum creatinine = 5.2 mg/dL (eGFR = 9 mL/min/1.73 m^2^), hemoglobin = 8.7 g/dL, serum urea = 92 mg/dL, and kalium = 4.7 mmol/L. The general condition of the patient was significantly improved, and she was instructed to continue the urine acidification therapy together with oral iron therapy.

### 2.5. Follow-Up

On the sixth week follow-up visit, the general condition of the patient was good, with a complete cessation of the gross hematuria. The hemoglobin level was 9.5 g/dL and kalium level was 5.2 mmol/L; however, renal function did not improve (serum creatinine level was 5.6 mg/dL, eGFR = 8 mL/min/1.73 m^2^), and consequently, we extracted the double-J stent (which had minimal calcifications at its proximal end) since we considered that it did not cause any improvement in the patient’s renal function. Urine pH was 6.0, and the special CBEU urine culture for *Corynebacterium urealyticum* was negative. The non-contrast abdominal CT scan showed partial regression of the encrustations, and the maximum density of the calcifications had decreased to 390 Hounsfield units. Considering these improvements, oral acidification of the urine with L-methionine was continued (Figure 4).

Unfortunately, the slow progression of the chronic kidney disease led to the indication of renal replacement therapy, thus prompting the patient to finally start chronic hemodialysis after eleven months.

## 3. Discussion

Encrusted pyelitis is an underdiagnosed, severe, and rare infectious disease that affects the urothelium, mostly caused by *Corynebacterium urealyticum,* which is usually missed in routine urine cultures. [1,2]. The identification of *C. urealyticum* is one of the biggest challenges for bacteriologists because it does not grow well after standard incubation [3]. *C. urealyticum* can only be isolated after 48 h of incubation at 35–37 °C, preferably in 10% CO_2_, mainly on blood agar [4]. The majority of *C. urealyticum* strains are multi-drug resistant, with universal susceptibility to vancomycin and teicoplanin [3]. *C. urealyticum* is a non-hemolytic, Gram-positive, aerobic bacillus with high urease activity, which is the cause of urine alkalinization (pH > 8) and magnesium ammonium phosphate stone formation (struvite). It has a marked tropism for urothelial cells, as demonstrated by Marty et al., which explains its ability to reach the upper urinary tract and produce encrustations in the collecting system [5]. A crystallographic analysis of the encrustations has revealed these to be predominantly struvite, but also apatite and proteins [6]. Urinary infections due to *C. urealyticum* are mostly nosocomial—the pathogen resides frequently in the groin of elderly patients receiving broad spectrum antibiotics, eventually favoring the colonization of urinary catheters and subsequent infections of the bladder [4]. The urinary tract infection requires a history of immunosuppression or prolonged antibiotic therapy, a history of urological procedures with contamination of the urinary tract, or a pre-existing inflammatory or neoplastic lesion of the urothelium, which provide favorable conditions for stone encrustation [1,7,8].

Encrusted uretero-pyelitis can appear in many clinical forms, and symptoms are not specific. Symptoms of cystitis are usually present and are similar to those produced by other bacteria. Patients frequently present with fever, dysuria, gross hematuria, and flank pain. Clinical aspects suggesting urinary tract infection with *Corynebacterium urealyticum* are the strong odor of ammonia in the urine, persistent alkaline pH, and the formation of struvite stones. Renal failure is usually present in patients with bilateral encrusted pyelitis, which involves native or transplanted kidneys. The diagnosis is based on a non-contrast, enhanced CT scan combined with urinalysis and a cytobacteriological examination of the urine (CBEU) in order to prove the infection with *C. urealyticum*. The effective accuracy of each imaging technique in detecting encrusted plaques is difficult to assess due to the rare occurrence of this disease. CT is the optimal imaging technique for the diagnosis of encrustations, particularly in the upper urinary tract, because it permits excellent visualization of the walls of the collecting system and can detect even thin radiolucent calcifications. A non-contrast, enhanced CT scan shows high-density edging lesions of the upper urinary tract that can be thin and regular or thick and irregular, and which can be differentiated from staghorn calculi through multiplanar reconstruction. It can also detect associated infectious complications such as perinephric abscess or periureteral inflammatory changes with fat stranding. Although contrast-enhanced CT could add valuable information regarding the functionality of the kidneys, thus documenting an eventual urinary tract obstruction, we recommend it with great care, especially in patients with altered renal function, due to the increased risk of contrast induced nephropathy. If endoscopic evaluation is performed, calcified plaques under the mucosa of the urinary tract can be observed. Frequent complications of encrusted pyelitis are obstructive uropathy associated with end-stage renal failure, ureteral stenosis, and renal abscesses [1,6,7,8,9].

In one macroscopic histological analysis of an excised kidney graft due to encrusted pyelitis, the thickened walls of the collecting system with superficial calcifications were revealed. In a microscopic analysis of the collecting system, three layers were revealed: a superficial layer with necrosis and calcifications, an intermediate layer with inflammatory changes, and a third layer corresponding to normal renal parenchyma [8].

Because of its slow growth (48–72 h), *C. urealyticum* is easily missed during normal CBEU, and therefore it is mandatory to communicate the suspicion of atypical bacteria to the bacteriologist. *C. urealyticum* can also be identified among stone fragments, plaques, or calcified mucosal debris. If cultures are negative and there is a major suspicion of *C. urealyticum*, PCR can be used for rapid diagnosis [10].

Treatment of encrusted pyelitis initially consisted in the surgical removal of the encrusted plaques, with poor results, mostly due to poor knowledge of the pathophysiology of the disease. Aguado et al. emphasized the importance of antibiotic treatment in association with urinary acidification [1,11]. *C. urealyticum* is a multi-drug resistant organism with intrinsic susceptibility to glycopeptides and 50% resistance rate to fluoroquinolones [2,3]. First-line antibiotic treatment consists of vancomycin or teicoplanin, whose efficacy is not influenced by the urinary pH. The calcified encrustations may be dissolved by oral or topical acidification. If encrustations are thin and not too extensive, oral acidification of the urine may be sufficient to dissolve the plaques [12,13]. When encrustations are thick and extensive, local acidification should be performed with urinary acidification solutions. Topical dissolution is performed via a nephrostomy tube and ureteral catheter in order to avoid high renal pressure, which must be maintained below 25 cm H_2_O. Thomas solution (sodium gluconate 27 g, citric acid 27 g, malic acid 27 g, distilled water 1000 mL) or hemiacridine solution are mostly used. Local flow rate must be adapted to patient tolerance (20–50 mL/h) [6]. The most frequent complication is candiduria. Oral dissolution of the plaques as a non-invasive alternative was first described by Khallouk et al. in 2006 and is achieved through the oral administration of ammonium chloride or L-methionine, with no significant side effects and good tolerability [8,12].

Combined antibiotic therapy and urine acidification must be administered for several weeks. Urine cultures and evaluation CT scans are performed at various intervals to assess treatment efficacy. Treatment duration is not well-established as it depends on the obtained results [6].

## 4. Literature Review

We performed an extensive literature search on PubMed for articles published up to December 2021, searching for cases of encrusted uretero-pyelitis caused by *Corynebacterium urealyticum*. No language restrictions were applied. Our search included 19 published case reports, with a total of 55 adult patients and 3 pediatric patients (aged 4, 9, and 13 years) (Table 1). Mean adult patient age was 65 years, 25 of them were males and 30 were females. The reports involved 40 patients (69%) with native kidneys and 18 patients (31%) with kidney grafts. At initial presentation and admission, 41 patients (71%) had renal failure, and 35 had obstructive uropathy (60%). Diagnosis of encrusted uretero-pyelitis caused by *C. urealyticum* was made mostly with the help of urine cultures on selective media and imaging of the urinary tract. Antibiotic therapy with vancomycin or teicoplanin was administered in 56 of the 58 reported cases (in two cases, the used antibiotic was not specified). Urine diversion was performed in 49 cases (84%) and consisted mostly of a combination of percutaneous nephrostomy and ureteral double-J catheter in order to ensure good urinary drainage and to facilitate continuous irrigation with urinary acidifying agents. Urinary acidification was achieved in 40 cases (69%), with continuous irrigation with Thomas solution through a nephrostomy tube. Chemolysis was well-tolerated overall, but it required strict supervision to prevent complications, to ensure that the urine remained sterile, and to maintain unobstructed inflow and outflow with low intrapelvic pressure. In four cases, oral acidification with L-methionine was used, with good results and partial recovery of renal function. Acetohydroxamic acid was used in association with continuous irrigation in two cases. Endoscopic surgical treatment, consisting of percutaneous nephrolithotripsy or ureteroscopy, was performed in 13 of the reported cases (22%), while open surgical treatment was performed in 4 cases (7%). Outcome was favorable in 34 of the cases reported (59%), with good recovery of eGFR and complete disappearance of the encrusted calcifications on the follow-up CT scan. Partial recovery of eGFR with partial improvement of the encrusted calcifications was noted in three cases. The progression of chronic kidney disease and the need for hemodialysis was observed in 13 patients (22%). One patient died of septic shock, two due hemorrhagic complications, and five due to other causes [9,11,13,14,15,16,17,18,19,20,21,22,23,24,25,26,27,28,29,30].

## 5. Conclusions

Encrusted uretero-pyelitis caused by *C. urealyticum* is an underdiagnosed, rare, and severe disease. Confirmation of the infection requires the use of a special cytobacteriological examination of the urine (CBEU) technique. The treatment is initially conservative and consists in a combination of specific antibiotic therapy and urinary acidification, administered either locally or orally. Oral acidification of the urine is a less invasive and efficient alternative to topical urinary acidification, with a lower risk of treatment-related complications. The follow-up regimen and the endourological treatment plan are not well-established and depend mostly on the results obtained.

In our case, the patient had a history of encrusted cystitis, with no initial identification of the infection with *C. urealyticum,* which led to the irreversible damage of the kidney function despite the partial improvement of the encrusted uretero-pelvic calcifications after antibiotherapy and urine acidification therapy.

## Figures and Tables

**Figure 1 diagnostics-12-02239-f001:**
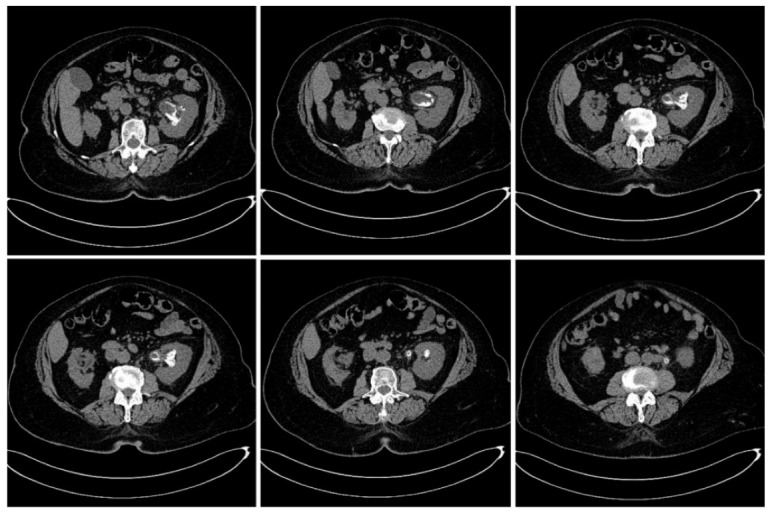
Axial plane images of abdominal and pelvic CT scan without contrast showing multiple pyelic, calyceal, and ureteral calcifications specific to encrusted uretero-pyelitis.

**Figure 2 diagnostics-12-02239-f002:**
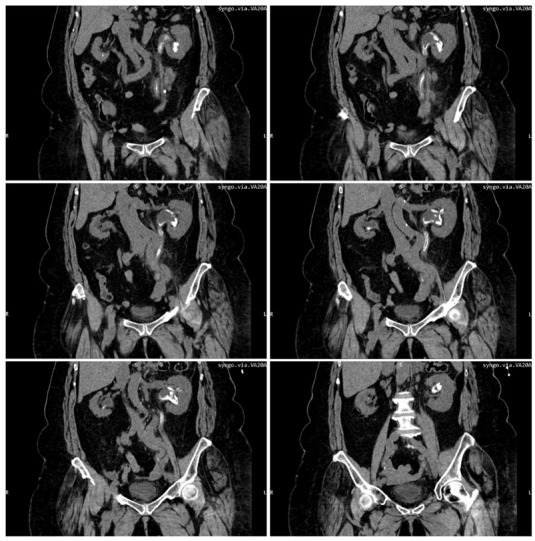
Multiplanar reconstruction of abdominal and pelvic CT scans without contrast shows multiple ureteral, pyelic, and calyceal encrustations.

**Figure 3 diagnostics-12-02239-f003:**
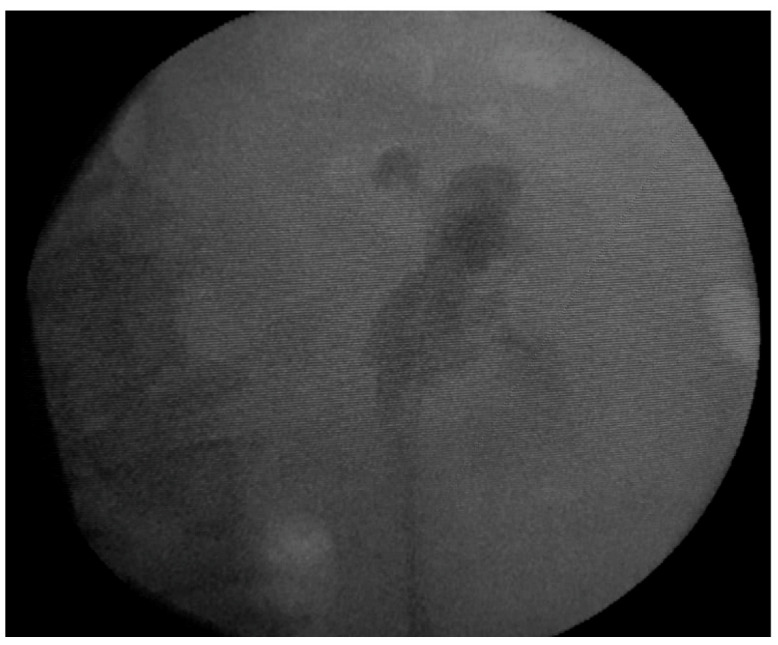
Fluoroscopic image after the placement of the left ureteric stent.

**Figure 4 diagnostics-12-02239-f004:**
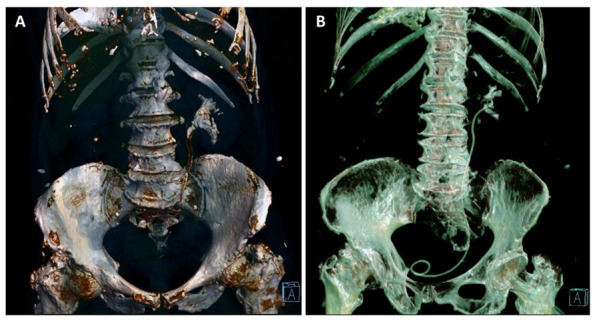
Three-dimensional reconstruction based on the non-contrast CT examinations of the patient at the moment of diagnosis (**A**) and on the sixth week follow-up visit (**B**), with the ureteric stent in place, showing the partial regression of the pyelic and calyceal calcifications, and the complete regression of the upper ureter calcifications.

**Table 1 diagnostics-12-02239-t001:** Published cases of encrusted uretero-pyelitis caused by *Corynebacterium urealyticum*.

Study	Number of Patients	Special Criteria	Renal Failure	Obstructive Uropathy	Antibiotic Therapy	Urinary Diversion	Urinary Acidification	Surgical Treatment	Outcome
J.M. Aguado et al., 1993, Spain [11]	7	Transplant	6	7	7	7	-	Unknown	Favorable = 4 Unfavorable = 3
B. Domínguez-Gil et al., 1999, Spain [14]	1 (female)	Transplant	1	1	1	1	-	1	Favorable = 1
P. Meria et al., 1999, France [13]	4 Male = 1 Female = 3	Transplant (1)	1	3	4	4	4	1	Favorable = 3 Unfavorable = 1
A. Hertig et al., 2000, France [15]	4 Male = 2 Female = 2	-	3	3	2	3	2	1	Death = 1 Favorable = 2 Unfavorable = 1
S. Giannakopoulos et al., 2001, Greece [16]	1 (male)	-	-	1	1	1	-	1	Favorable = 1
S. Garcia et al., 2004, Spain [17]	2	Transplant	1	2	2	1	-	1	Favorable = 1 Unfavorable = 1
P. Meria et al., 2004, France [18]	2 (male)	Pediatric (2) Transplant (1)	1	1	2	2	2	1	Favorable = 1 Unfavorable = 1
R. Lee et al., 2004, USA. [19]	1 (male)	Pediatric Transplant	-	-	1	1	1	1	Faborable
S. van Hooland et al., 2005, Belgium [20]	1 (male)	Neobladder	1	-	1	1	1	-	Partially favorable
S. Lieten et al., 2011, Belgium [21]	1 (male)	-	1	1	1	1	1	-	Death
N. Anagnostou et al., 2012, Australia [22]	1 (male)	-	-	-	1	-	1	1	Death
L. Cappuccino et al., 2014, Italy [9]	1 (male)	-	1	1	1	1	1	-	Favorable
R. Saljoghi et al., 2016, France [23]	1 (male)	-	1	1	1	1	1	-	Favorable
F.M. Sánchez-Martín et al., 2016, Spain [24]	12 Male = 2 Female = 10	-	8	Unknown	12	9	9	4	Favorable = 8 Partially favorable = 2 Unfavorable = 2
M. Vergura et al., 2018, Italy [25]	1 (male)	Autoimmune thrombocyto-penic purpura	1	1	1	1	1	-	Death
H. Sakhi et al., 2020, France [26]	15 Male = 6 Female = 9	Transplant (4) Lymphoma (1)	12	11	15	13	15	2	Favorable = 7 Unfavorable = 4 Death = 4
L. Sabiote et al., 2020, Spain [27]	1 (male)	-	1	1	1	1	1	1	Favorable
M. Johnson et al., 2020, USA. [28]	1 (female)	Transplant	1	1	1	1	-	1	Favorable
A. Loghmari et al., 2020, Tunisia [29]	1 (male)	-	-	1	1	-	-	1	Favorable

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
