# Peer review of "Encrusted Uretero-Pyelitis Caused by Corynebacterium urealyticum: Case Report and Literature Review"

_diagnostics, 2022, doi:10.3390/diagnostics12092239_

Round 1
Reviewer 1 Report
Dear Authors, thank you for giving me the opportunity to review your valuable manuscript. There are some comments that I would like to make with the intention of improving it:
1. Please, add references to the statement about treatment of EUP (Lines 38, 182)
2. Provide the images to show regression of the encructations (Lines 115-117)
3. Specify the year of the Khallouk oral dissolution approach (Line 192)
4. Please, change the sentence so to make it clear there were 19 reported cases only with a total of 58 patients, out of which 3 were of pediatric age (specify the age of children, as it may reflect the velocity of the encrustation process) (Lines 201-203)
5. Please, specify antimicrobial therapy of the remainder 2 reported cases (Line 209)
6. Table 1: Add the columns “Year”, “Location/Country”. Add the drugs used, duration and dosages to the Columns “Antibiotic therapy” and “Urinary acidification”. It would be also valuable to give the ranges of urine pH of patients (if possible).
7. Please change “antibiotherapy” to “antibiotic therapy” or “antimicrobial therapy” throughout the text (Lines 21, 37, 97, 99, 194, 208, Table 1, 231)
8. Please change “sensitivity” to “susceptibility” throughout the text (Lines 90, 91, 129, 180)
Minor Comments:
Line 34: hospital-acquired (add hyphen)
Line 97: ciprofloxacin from the small letter
Line 99, 208: vancomycin from the small letter
Line 128, 208: urealyticum from the small letter
Line 181: change “constant” to “intrinsic
Line 182: correct to fluoroquinolones
Line 192: missing dot after Khallouk et al.
Line 197: Put the reference prior to the dot
Line 226: Please, italicize Corynebacterium urealyticum
Reviewer 2 Report
The author presented a 70-year-old female patient with solitary left functioning kidney and encrusted uretero-pyelitis caused by Corynebacterium urealyticum, which was treated by antibiotherapy and oral acidification with L-methionine. From literature review, encrusted uretero-pyelitis caused by C. urealyticum is an underdiagnosed, rare, and severe disease. Confirmation of the infection requires a special cytobacteriological examination of the urine (CBEU) technique. The treatment is initially conservative and consists in a combination of specific antibiotherapy and urinary acidification. The article is well written and provides comprehensive knowledge to the readers.
I have some minor questions:
1.Did the patient have recurrent UTI with same pathogen before? Is there possible colonization?
2.From the EUCAST data, the pathogen of this case was sensitive to fluoroquinolones. Why did you shift Ciprofloxacin to Vancomycin? Is there clinical unstable signs?
Reviewer 3 Report
A rare disease is reported. The presentation is excellent, and have a high quality to be published on this journal.
Author Response
Dear reviewer, thank you very much for your remarks. We have tried to further improve the article.